# MEDAKA: CONSTRUCTION OF BIOMEDICAL KNOWLEDGE GRAPHS USING LARGE LANGUAGE MODELS

## ABSTRACT

Knowledge graphs (KGs) are increasingly used to represent biomedical information in structured, interpretable formats. However, existing biomedical KGs often focus narrowly on molecular interactions or adverse events, overlooking the rich data found in drug leaflets. In this work, we present (1) a *hackable*, end-to-end pipeline to create KGs from unstructured online content using a web scraper and an LLM; and (2) a curated dataset, MEDAKA, generated by applying this method to publicly available drug leaflets. The dataset captures clinically relevant attributes such as side effects, warnings, contraindications, ingredients, dosage guidelines, storage instructions, and physical characteristics. We evaluate it through manual inspection and with an *LLM-as-a-judge* framework, and compare its coverage with existing biomedical KGs and databases. We expect MEDAKA to support tasks such as patient safety monitoring and drug recommendation. The pipeline can also be used for constructing KGs from unstructured texts in other domains. Code and dataset are available at `github.com/medaka25/medaka-25`.

## 1 INTRODUCTION

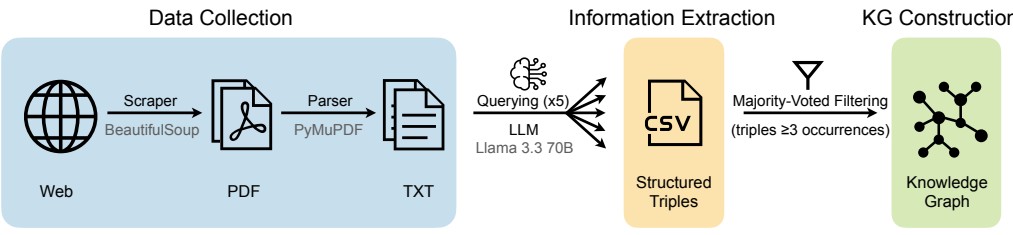

Figure 1: Overview of the proposed pipeline for constructing the MEDAKA KG.

The biomedical domain faces rapid growth in data from sources such as drug leaflets, clinical notes, scientific articles, and electronic health records. Much of this information is unstructured, making it difficult for machines to parse. By organizing this data into a structured KG, biomedical knowledge can be represented as a network of nodes and relations.

Biomedical KGs integrate heterogeneous sources and support tasks such as drug repurposing (Doshi & Chepuri, 2022; Gogineni, 2022; Huang et al., 2024), detection of adverse drug reactions via polypharmacy modeling (Zitnik et al., 2018), drug discovery (Zeng et al., 2022), and personalized treatment planning (Gyrard et al., 2018). Recent surveys highlight the growing importance of biomedical KGs and using LLMs for constructing them, as a promising direction in medical research (Cui et al., 2025).

Despite these advantages, building and maintaining high-quality KGs is expensive, slow, and labor-intensive, requiring significant manual effort and domain expertise. Traditional knowledge bases

often struggle to keep up with the pace of new information in this domain and are costly to maintain, limiting their scale, scope, and suitability for real-time applications.

Advances in LLMs offer an effective alternative for constructing KGs across domains (Venkatakrishnan et al., 2024; Zhu et al., 2024). Trained on large corpora, LLMs can comprehend complex biomedical text, extract entities, and infer relations, providing a scalable way to generate structured knowledge from unstructured sources while reducing manual annotation and rule engineering. Applications include prescription label interpretation for patient medication management (Thetbanthad et al., 2025) and state-of-the-art medication information extraction from clinical text (Richter-Pechanski et al., 2025).

**Contributions.** In this work, we present:

1. An end-to-end pipeline for automatically constructing KGs from unstructured text using a web scraper and an LLM.
2. MEDAKA, a KG built from publicly available drug leaflets sourced online and released as a downloadable CSV.

The method is modular, reproducible, and easily adaptable to other domains. Both the LLM and the input data can be substituted to generate different KGs. The resulting KG, MEDAKA, captures relevant drug-related information such as active and inactive ingredients, warnings, dosage guidelines, and storage requirements–attributes often missing in existing medical databases such as SIDER (Kuhn et al., 2016) and the FDA Adverse Event Reporting System (FAERS) (U.S. Food and Drug Administration, 2024).

## 2 PROPOSED METHOD

In this section, we describe the pipeline used to construct MEDAKA . This includes collecting raw data, extracting structured information using an LLM and organizing the resulting nodes and labeled edges into a structured KG.

**Overview.** The proposed pipeline constructs a biomedical KG from unstructured drug leaflets. It includes three main stages: (1) data collection via scraping and parsing; (2) LLM-based information extraction; and (3) KG construction. Figure 1 illustrates each step in the pipeline, with each transition corresponding to a processing action described in the following subsections.

**Graph schema.** A KG is a directed graph with labeled edges. The schema of MEDAKA defines key biomedical node types, including `Drug`, `ActiveIngredient`, `InactiveIngredient`, `SideEffect`, `Warning`, `Contraindication`, `DosageInfo`, `StorageCondition`, `Color`, and `Shape`. These nodes are connected by relations such as `hassideeffect`, `haswarning`, `hascontraindication`, `hasactiveingredient`, `hasinactiveingredient`, `hasdosageinfo`, `hasstorageinfo`, `hascolor`, and `hasshape`, forming a structured biomedical KG.

**Data collection via web scraping.** In this work, we use publicly available drug leaflets sourced from the Health Products Regulatory Authority (HPRA), Ireland's national agency responsible for regulating medicines. Pharmaceutical companies draft documents such as the Summary of Product Characteristics and Package Information Leaflets as part of the drug approval process. The HPRA reviews these documents to ensure they meet standards of safety, quality, and efficacy before granting marketing authorization. These standardized documents, intended for healthcare professionals and patients, are publicly accessible via the HPRA website[1] and serve as reliable sources of biomedical information.

We collected approximately 13,000 drug leaflets related to human medicine with a lightweight web scraping script built with the Python package `BeautifulSoup` (Richardson, 2023). The script parsed the HTML source of relevant HPRA web pages to extract direct PDF download links of the leaflets, which were then stored locally.

---

[1] https://www.hpra.ie

To validate our approach, we also tested the scraper on a second online pharmacy website. Unlike the HPRA site, this platform relied heavily on JavaScript to generate content and imposed restrictive crawling rules, rendering `BeautifulSoup` ineffective. In response, we experimented with more advanced tools such as ScraperAPI, Crawl4AI, and Selenium, which support dynamic content rendering and could successfully access the data.

**PDF parsing.** The collected drug leaflets were parsed using the `PyMuPDF` library (Inc. & Contributors, 2025) to extract their full textual content. Each leaflet was converted into a plain text string and served as input to a prompt-based LLM pipeline, where it was combined with predefined prompts for information extraction.

**Information extraction using LLMs.** To transform the parsed leaflet text into a structured representation, we developed a prompt-based LLM pipeline. The pipeline accepts the full plain-text leaflet as input with predefined prompts, allowing single-pass processing that preserves context and reduces redundancy. The model outputs subject–relation–object triples, stored in a CSV file for KG construction.

LLMs have a finite input size, referred to as the context window. Drug leaflets typically contain 4,000–8,000 words, often exceeding the context window of smaller models such as Mistral 7B (Jiang et al., 2023a). Segmenting documents caused repetition or omissions, so we used the LLaMA 3.3 70B Instruct model (Grattafiori et al., 2024), which supports longer context windows and enabled complete single-pass processing. In preliminary experiments, we tested Mistral 7B and Qwen-32B (Yang et al., 2025) on 200 leaflets. Qwen produced overly verbose outputs, while Mistral required chunking that introduced redundancy. LLaMA consistently yielded concise and complete extractions, and was therefore selected as our final model.

We then refined the prompt through several iterations to improve stability and reduce redundancy. Consistency was reinforced with in-context learning, embedding illustrative examples of triples directly in the prompt. The final prompt design targeted clinically relevant attributes such as drug name, side effects, ingredients, warnings, contraindications, dosage guidelines, storage instructions, and physical characteristics, with the drug name specified as the subject and relations restricted to the nine defined types.

To enhance reliability and mitigate erroneous outputs arising from LLM hallucinations, we used a majority voting strategy. Each leaflet was processed by submitting the same prompt to the LLM five times, yielding five independent outputs. For every triple across the five outputs, we recorded its frequency of occurrence across the generations and computed a confidence score, defined as the ratio of its frequency to five.

**KG construction.** Following the LLM-based extraction process, we applied a post-processing step to construct the final KG. Triples were filtered using majority voting, retaining only those with a confidence score of at least 0.5 (i.e., present in three or more generations). To further reduce redundancy and enforce consistency, all triples were normalized by converting subjects, relations, and objects to lowercase. The validated and normalized triples collectively form MEDAKA.

## 3 DATASET DESCRIPTION

As a demonstration of our pipeline for constructing biomedical KGs, we created the dataset MEDAKA as a case study from drug leaflets. It captures connections between `Drug` entities and various biomedical concepts, such as `SideEffect`, `Warning`, `Contraindication`, `ActiveIngredient`, `InactiveIngredient`, `Color`, `Shape`, `StorageInfo`, and `DosageInfo`. Figure 2 shows the distribution of entities and relations in MEDAKA, while Figure 3 presents subgraphs centered on two example drugs, Toltertan SR and Catasart Plus, illustrating their connections to key biomedical entities.

MEDAKA consists of 41,142 nodes and 466,359 directed, labeled edges. Figure 4 shows the degree distribution for three core node types–`Drug`, `SideEffect`, and `Contraindication`, where the x-axis indicates the number of neighbors an entity has, and the y-axis shows how many entities have that many neighbors. Figure 5 visualizes the clustering of drug nodes based on Jaccard similarity computed over binary vectors of their connected biomedical entities, grouping together drugs

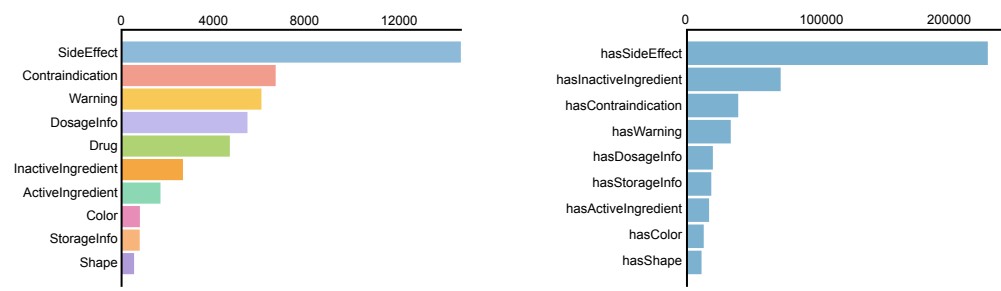

Figure 2: Distribution of entities (left) and relations (right) in MEDAKA.

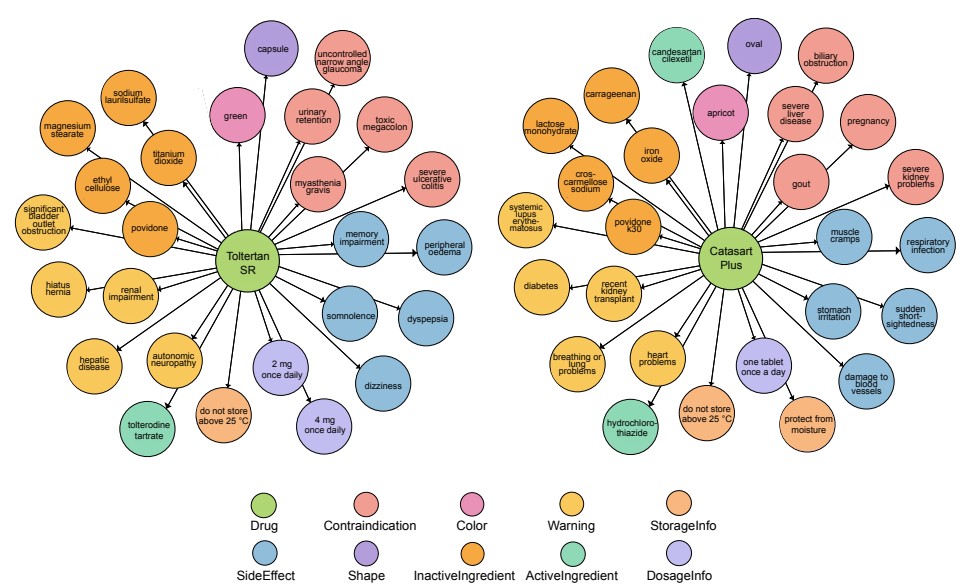

Figure 3: Subgraphs from MEDAKA centered on the drugs *Toltertan SR* and *Catasart Plus*, showing connections to different biomedical entities. Only a subset of nodes is displayed for clarity.

with similar sets of connected biomedical entities. MEDAKA is a well-organized, semantically rich dataset that has the potential to support tasks such as drug recommendation and patient safety alerts.

## 4 EVALUATION

Evaluating the quality of automatically constructed KGs is a challenge, especially in domains without standard benchmark datasets. The complexity of KGs makes evaluation particularly difficult. To evaluate our dataset, we employed three strategies: (i) human evaluation of randomly sampled triples, (ii) validation with an LLM-as-a-judge to assess scalability and consistency, and (iii) coverage comparison with existing biomedical KGs and databases. Since no reference KG exists for this domain, these methods together provide a comprehensive assessment of MEDAKA.

**Human evaluation.** We randomly sampled 100 drug leaflets from the dataset, corresponding to 3,549 extracted relations. The distribution of relation types in this subset was closely aligned with the full dataset, as confirmed by 95% confidence intervals around the sample proportions. Each triple was manually annotated as Correct (True Positive), Incorrect (False Positive), or Partially Correct (Incomplete). The evaluation yielded 3,427 correct triples (96.6%), 34 incorrect triples (0.9%), and 88 partially correct triples (2.5%). To additionally estimate recall, we manually inspected 10 randomly selected leaflets (619 relations) for false negatives, to identify relations present in the

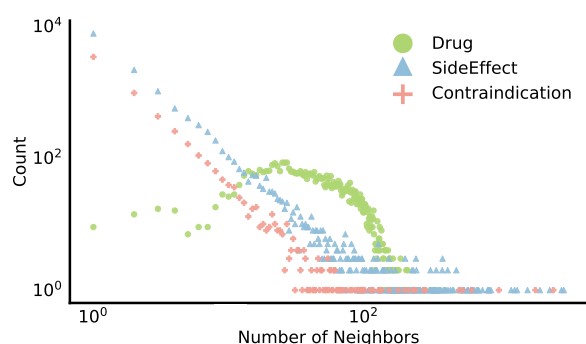

Figure 4: Degree distributions of three core node types in MEDAKA: `Drug`, `SideEffect`, and `Contraindication`.

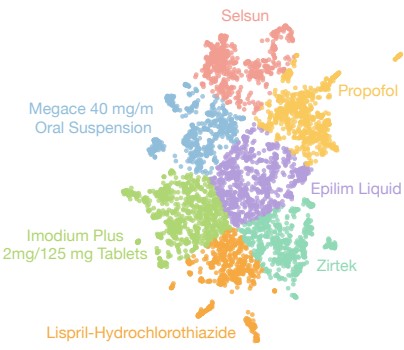

Figure 5: Clustering of drug nodes in MEDAKA based on Jaccard similarity of connected biomedical entities. Each cluster groups drugs with similar attributes, with the representative drug shown as the label.

text but not extracted into the KG. We identified 79 false negatives, corresponding to a recall of 87.2% across the evaluated sample. Due to the high cost of exhaustive manual validation, recall was assessed on a subset of 10 leaflets.

**LLM-as-a-judge evaluation.** We further assessed the 3,549 triples using an *LLM-as-a-judge* framework with the gpt-oss-120b model (Agarwal et al., 2025), which is well suited for reasoning tasks. Each leaflet, together with its extracted triples, was provided to the model, which then assigned one of three labels, Correct (True Positive), Incorrect (False Positive), or Partially Correct (Incomplete), and produced a short justification for its choice. The evaluation prompt was iteratively refined to ensure reliable judgments, and in-context learning was applied by providing examples of correct, incorrect, and partially correct triples.

For instance, the triple (*zitrease allergy*, `hassideeffect`, *somnolence*) was labeled *Correct* with the reasoning that "somnolence is a common side effect." The triple (*oxynorm*, `hasinactiveingredient`, *iron*) was judged *Partially Correct* because the leaflet listed *iron oxide* as an excipient rather than *iron*, and the model correctly identified this distinction. Out of 3,549 triples, the LLM judged 3,439 (96.9%) as Correct, 24 (0.7%) as Incorrect, and 86 (2.4%) as Partially Correct, closely matching human labels (Table 1). This close alignment demonstrates strong agreement with human evaluation, suggesting that LLM-based judgment can serve as an efficient alternative for large-scale quality assessment while human evaluation remains the reference standard.

**Coverage comparison.** To evaluate the coverage of MEDAKA relative to established biomedical resources, we performed a comparison of the types of information it captures with those available in widely used databases, including DrugBank (Knox et al., 2024), SIDER (Kuhn et al., 2016), and

Table 1: Comparison of human and LLM-as-a-judge evaluations on randomly sampled triples.

| Category | Human evaluation | | LLM evaluation | |
|---|---|---|---|---|
| | Count | % | Count | % |
| Correct (True Positive) | 3427 | 96.6 | 3439 | 96.9 |
| Incorrect (False Positive) | 34 | 0.9 | 24 | 0.7 |
| Partially Correct (Incomplete) | 88 | 2.5 | 86 | 2.4 |
| **Total** | **3549** | **100.0** | **3549** | **100.0** |

FAERS (U.S. Food and Drug Administration, 2024). Although these databases provide valuable information on drug–target interactions, side effects, and adverse event reports, they often lack many of the contextual and real-world usage details that are present in drug leaflets. As shown in Table 2, MEDAKA captures additional attributes such as contraindications, warnings, storage conditions, and physical characteristics of drugs.

## 5 RELATED WORK

**Biomedical KGs from heterogeneous sources.** Several large-scale biomedical KGs have been developed by integrating diverse sources. Examples include iKraph, which combines PubMed abstracts with 40 public databases (Zhang et al., 2025), PrimeKG, which harmonizes 20 sources via UMLS (Chandak et al., 2023), and SPOKE, which unifies 41 databases into 27 million nodes using ontology-driven integration (Morris et al., 2023). Other efforts, such as DRKG (Ioannidis et al., 2020) and DrugMechDB (Gonzalez-Cavazos et al., 2023), focus on drug–disease and drug–mechanism relationships to support tasks like drug repositioning. More specialized initiatives, including disease-focused KGs constructed from PubMed abstracts with MONDO-based standardization (Thaokar et al., 2024) and BioKGrapher, which builds condition-specific graphs using NER, entity linking, and ontology alignment (Schäfer et al., 2024), show efforts in automating text-driven KG construction. Collectively, these works highlight the importance of integrating diverse sources to build comprehensive biomedical KGs.

**LLM-based KGs.** Recent work shows that LLMs are increasingly used for biomedical KG construction. A key focus has been relation extraction, explored with GPT-based models (Zhang et al., 2024) and ensemble methods that use attention for robustness (Jia et al., 2024). Beyond this, LLMs support end-to-end pipelines for entity and relation extraction from clinical records (Arsenyan et al., 2023), as well as interactive tools such as MedKG, which uses ChatGPT to refine graphs from textual data (Jiang et al., 2023b). More advanced methods extend these capabilities, including TxGNN, which leverages medical KG data for therapeutic candidate ranking (Huang et al., 2024), MedGraphRAG, which combines graph-based retrieval with LLMs to produce evidence-grounded responses (Wu et al., 2024), BioStrataKG, which applies LLMs for cross-document retrieval and reasoning (Feng et al., 2025), and AMG-RAG, which integrates LLMs with external evidence to enable continuous KG updates (Rezaei et al., 2025). Together, these approaches show the growing role of LLMs in both the construction and application of biomedical KGs.

**Structured drug databases.** In addition to KGs, structured drug databases provide curated information. SIDER (Kuhn et al., 2016) compiles adverse drug reactions from public sources and drug labels but is limited to drug–side-effect pairs, omitting important contextual attributes such as contraindications and dosage. FAERS (U.S. Food and Drug Administration, 2024) collects real-time adverse event reports from healthcare providers and patients but is affected by noise, reporting bias, and the lack of structured, comprehensive drug profiles.

**Bridging biomedical KGs with real-world drug knowledge.** Most biomedical KGs focus on molecular and biological data while overlooking prescription-level details needed in clinical settings. For instance, PrimeKG (Chandak et al., 2023) models drug—disease associations through genes

Table 2: Comparison of existing biomedical KGs and databases against MEDAKA.

| Feature | iKraph | PrimeKG | DRKG | DrugMechDB | SIDER | FAERS | MEDAKA |
|---|---|---|---|---|---|---|---|
| Source | PubMed, public DBs | Clinical DBs | Clinical DBs | Curated mechanisms | Drug labels | Adverse event reports | Drug leaflets |
| Number of nodes | 10,686,927 | 129,375 | 97,238 | 32,588 | 7,894 | 113,115,568 | 41,142 |
| Drug entities | Yes | Yes | Yes | Yes | Yes | Yes | Yes |
| Warnings | No | No | No | No | No | No | Yes |
| Contraindications | No | No | No | No | No | No | Yes |
| Side effects | No | No | Yes | No | Yes | Yes | Yes |
| Active ingredients | No | No | No | No | No | Yes | Yes |
| Inactive ingredients | No | No | No | No | No | No | Yes |
| Dosage information | No | No | No | No | No | Yes | Yes |
| Storage instructions | No | No | No | No | No | No | Yes |
| Physical attributes | No | No | No | No | No | No | Yes |
| Extraction method | NLP pipeline | Data integration | Data integration | Manual curation | Manual extraction | Report compilation | LLM-assisted extraction |

and proteins and includes pharmacological data, but omits visual attributes such as shape and color, as well as warnings, dosage, and storage information. Similarly, DRKG (Ioannidis et al., 2020) is designed for molecular tasks such as drug repurposing but excludes patient-facing details. To address this gap, we present MEDAKA, a KG constructed directly from drug leaflets. It captures real-world, drug-centric attributes that are typically absent from existing KGs. Table 2 compares the coverage of MEDAKA with other biomedical KGs and databases.

# 6 CONCLUSION

In this work, we introduced MEDAKA, a large-scale biomedical KG derived from drug leaflets, alongside a modular pipeline for extracting structured knowledge from unstructured text. By integrating majority voting, systematic post-processing, and an *LLM-as-a-judge* framework, the pipeline produces high-quality triples that align closely with human annotations while capturing a broad spectrum of prescription-level drug information, including side effects, contraindications, warnings, dosage, storage, and ingredient details. While promising, the approach is constrained by the reliability of drug leaflets, the incompleteness of information available online, and variability in website structure that can occasionally affect web scraping. Future extensions could include enriching MEDAKA with disease-related information such as symptoms, risk factors, and definitions from external biomedical sources to enable more advanced reasoning on drug–disease relationships. Extending the pipeline to multilingual corpora would improve its global relevance, and its modular design makes it applicable for building KGs in other domains beyond biomedicine.

## ETHICS STATEMENT

In this work, we used publicly available drug leaflets released by the HPRA, which do not involve personal or patient data. The documents are openly licensed for reuse, provided the source is acknowledged and the content is reproduced accurately. We adhered to the ICLR Code of Ethics throughout this work.

## REPRODUCIBILITY STATEMENT

We provide the full pipeline implementation, including data collection, LLM-based extraction, KG construction, and evaluation scripts, in our anonymized GitHub repository[2]. The MEDAKA dataset is publicly released, with construction details described in the Proposed Method section. These resources enable full reproduction of our experiments and results.

## LLM USAGE

In our work, we used LLMs only in a minor way for language refinement and LaTeX corrections. They were not used for research ideation.

---

[2]https://github.com/medaka25/medaka-25

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
