# OpenReview forum: "Medaka: Construction of Biomedical Knowledge Graphs Using Large Language Models"
_ICLR.cc/2026/Conference — ICLR 2026 Conference Withdrawn Submission_

### Official Review · Reviewer_mSuv · 2025-10-29

**Soundness:** 1
**Presentation:** 2
**Contribution:** 1
**Rating:** 2
**Confidence:** 4

**Summary:**

The paper tackles the problem of KG construction in the biomedical field.
While this is a very interesting problem, this work is not mature in its current stage.
Here are the main points that I would suggest the authors focus on:
- When building a Knowledge Graph is very important to specify which entities are involved, why they have been considered, and what the use case is that we are actively trying to solve. I would suggest looking into the literature of knowledge engineering, ontology engineering, etc.
- especially in the biomedical field, there exists a myriad of well-established KG - I suggest the authors to explore https://bioportal.bioontology.org/
- When creating data resources to be reused, it would be advisable to use data standards that are commonly used for exchange, that define the schema, the provenance, etc.
- You might find this paper useful: https://www.nature.com/articles/sdata201618

**Strengths:**

- interesting problem

**Weaknesses:**

- When building a Knowledge Graph is very important to specify which entities are involved, why they have been considered, and what the use case is that we are actively trying to solve. I would suggest looking into the literature of knowledge engineering, ontology engineering, etc.
- especially in the biomedical field, there exists a myriad of well-established KG - I suggest the authors to explore https://bioportal.bioontology.org/
- When creating data resources to be reused, it would be advisable to use data standards that are commonly used for exchange, that define the schema, the provenance, etc.

**Questions:**

- What is the specific use case?
- Has the resource helped/improved any downstream task?

---

### Official Review · Reviewer_r8z8 · 2025-10-30

**Soundness:** 3
**Presentation:** 2
**Contribution:** 2
**Rating:** 4
**Confidence:** 5

**Summary:**

The paper introduces MEDAKA, a biomedical knowledge graph (KG) automatically generated from public drug leaflets using large language models (LLMs). The authors propose an end-to-end pipeline involving web scraping, PDF parsing, and LLM-based triple extraction (using LLaMA 3.3 70B). The dataset contains roughly 41 K nodes and 466 K edges, representing relations between drugs and biomedical attributes such as side effects, warnings, and dosage. Evaluation combines manual checking of a small subset (100 leaflets) and an LLM-as-a-judge setup, reporting ~97 % correctness and ~87 % recall.
The paper’s strengths lie in its clear design and reproducibility, but it lacks methodological novelty and thorough empirical analysis. The results show promise, yet the contribution is mostly applied and incremental, rather than conceptual or algorithmic.

**Strengths:**

The topic—automated biomedical KG construction—is timely and relevant. Presents a fully automated, reproducible pipeline that effectively integrates web scraping, LLM extraction, and KG construction. Combining human and LLM-based evaluation provides an interesting dual-validation approach.

**Weaknesses:**

The paper exhibits limited novelty, as it primarily integrates existing components such as web scraping, LLM prompting, and filtering without introducing new methodological insights. The evaluation is weak, lacking comparisons against baseline extractors or alternative LLMs. Moreover, the manual validation sample size is small, which makes the reported precision and recall less reliable. The work also omits downstream studies that could demonstrate practical utility, such as reasoning or drug-safety prediction tasks. Additionally, the dataset is geographically narrow—restricted to HPRA leaflets—which raises concerns about its generalizability to non-English or international data sources. Finally, the paper does not provide sufficient analysis of failure cases or error categories, leaving uncertainty about the robustness and limitations of the proposed pipeline.

**Questions:**

1.How consistent are extraction results across leaflets of different structure and length?
2.Could smaller open-source models achieve similar accuracy with lower cost?
3.What kinds of errors remain after majority voting—semantic, lexical, or relational?
4.Have you evaluated on non-HPRA or non-English documents to test generalization?
5.Is there a plan to align MEDAKA entities with biomedical ontologies (UMLS, SNOMED)?
6.How sensitive is the KG quality to prompt design or temperature settings?
7.Could retrieval-augmented prompting or agent further reduce hallucination?

---

### Official Review · Reviewer_vGdn · 2025-10-31

**Soundness:** 3
**Presentation:** 2
**Contribution:** 2
**Rating:** 2
**Confidence:** 3

**Summary:**

The paper introduces MEDAKA, a pipeline and dataset designed to construct biomedical knowledge graphs (KGs) from unstructured drug leaflets using web scraping and large language models (LLMs). The approach extracts clinically relevant information such as side effects, contraindications, dosage, and storage instructions, and organizes it into a structured format. The authors present a comprehensive comparative table summarizing related work and evaluate the generated dataset through both manual annotation and an LLM-as-a-judge framework, reporting high agreement between methods. The paper also provides public access to the dataset and code.

**Strengths:**

1. Overall, the paper is clearly written and easy to follow. The authors present their ideas in a logical and coherent manner, and the main message of the work is well-articulated throughout the manuscript.

2. The inclusion of a comprehensive comparative table (Table 2) is particularly valuable, as it effectively summarizes key aspects of related work and allows readers to clearly see how the proposed approach differs from and improves upon existing methods.

3. The evaluation of the generated dataset against both manual annotations and an LLM-as-a-judge framework is a strong aspect of the paper. The high agreement scores reported between these evaluation methods support the reliability and quality of the dataset, reinforcing confidence in the robustness of the proposed approach.

**Weaknesses:**

- The authors do not appear to normalize entities in their dataset. Upon inspection, I found examples such as "dizziness" and "feeling dizzy", or "anxiety" and "feeling anxious", which should be normalized to a single entity ID representing a unique concept. Ideally, normalization should leverage standardized biomedical dictionaries such as MeSH, organized within ontologies like MedDRA and UMLS. For example, the "dizzy" concept corresponds to the MeSH term D004244, and "anxiety" to the SNMI code F-92238. This would allow the generated knowledge graph to be easily mapped to other biomedical ontologies, such as PrimeKG. Using only surface forms, as is currently done, prevents such integration and may introduce inconsistencies (e.g., treating "dizziness" and "feeling dizzy" as distinct entities). Similarly, standardized resources such as RxNorm and DrugBank should be used to canonicalize drug names.
- Similar to the previous point, I would also suggest leveraging established ontologies such as SIDER, DrugBank, and FAERS for relation types like "has side effect." Ideally, these relation types should be aligned with standardized biomedical knowledge graph schemas. At present, it is not clear how relation types are defined within Medaka.
- The authors should expand the related work section to include a discussion of closely related datasets, such as:
	- DailyMed SPL (https://dailymed.nlm.nih.gov/dailymed/spl-resources.cfm)
	- Demner-Fushman, Dina, et al. "A dataset of 200 structured product labels annotated for adverse drug reactions." Scientific data 5.1 (2018): 1-8.
	- Kass-Hout, Taha A., et al. "OpenFDA: an innovative platform providing access to a wealth of FDA’s publicly available data." Journal of the American Medical Informatics Association 23.3 (2016): 596-600.
- The authors do not provide prompts used to construct KG, making it hard to reproduce. The prompts should be described in appendix.
- The authors do not compare with existing approaches to generate KG from textual sources using LLMs, such as:
	- Yang, Hao, et al. "Large Language Model–Driven Knowledge Graph Construction in Sepsis Care Using Multicenter Clinical Databases: Development and Usability Study." Journal of Medical Internet Research 27 (2025): e65537.
	- Edge, Darren, et al. "From local to global: A graph rag approach to query-focused summarization." arXiv preprint arXiv:2404.16130 (2024).
	- Chen, Hanzhu, et al. "SAC-KG: Exploiting Large Language Models as Skilled Automatic Constructors for Domain Knowledge Graph." Proceedings of the 62nd Annual Meeting of the Association for Computational Linguistics (Volume 1: Long Papers). 2024.
- The related work section is divided into multiple paragraphs, each addressing a different type of related research. However, the authors do not clearly explain how their work compares to the studies discussed in each paragraph, nor do they highlight which specific gaps their approach addresses. As a result, the paper's contribution is difficult to identify.

**Questions:**

1. Could the authors clarify what is meant by “end-to-end” in the abstract? It would be helpful to specify which components of the pipeline are automated and how this relates to the main contributions of the work.

2. Was any form of few-shot learning used in the pipeline — for example, providing the model with representative examples before generating triples from text? If so, how does this affect the quality or consistency of the extracted relations?

3. It is not clear how the relation types were defined. Were they extracted directly from the text, or was any normalization or mapping process applied to align them with standardized schemas?

---

### Official Review · Reviewer_1LXa · 2025-10-31

**Soundness:** 2
**Presentation:** 3
**Contribution:** 2
**Rating:** 2
**Confidence:** 2

**Summary:**

The paper introduces MEDAKA, a large-scale biomedical knowledge graph automatically constructed from drug leaflets using an LLM-based pipeline. The system handles ~13k publicly available HPRA drug leaflets, parses PDFs, and extracts subject-relation-object triples via LLMs, followed by majority vote to enhance quality. The generated graph covers 9 relation types that are clinical insightful. Evaluation combines human annotation, LLM-as-a-judge consistency checking, and coverage comparison against some know benchmarks.

**Strengths:**

1. The paper designs an end-to-end pipeline for KG building, which is pretty practical and could benefits both clinical and AI researchers.

2. Mining on drug leaflets looks novel to me, which could serve as a meaningful addition to existing biomedical KGs.

**Weaknesses:**

1. The paper as limited technical contribution but integrates existing works to an engineering pipeline. I feel that core value lies in data source rather than ML methodological advances.

2. Human efforts cover only 100 leaflets with the recall estimated on 10. The dataset's scale is far more bigger than the manual coverage, so 96 % accuracy may be optimistic.

3. The paper uses Llama series for extraction. I'm unsure if this is unbiased but I don't see ablation studies.

4. The authors run 5 times per leaflet using LLMs but the dataset is huge. I cannot see that the method is scalable.

5. There are too few quantitative analysis in the experiments session. The results also lack statistical significance.

**Questions:**

Please refer to my comments in weaknesses.

---

### Note · Authors · 2025-11-17

**Comment:**

We thank the reviewers for their helpful comments and for pointing us to several useful references.

**Withdrawal Confirmation:**

I have read and agree with the venue's withdrawal policy on behalf of myself and my co-authors.